# Longitudinal Strain Analysis and Correlation with TIMI Frame Count in Patients with Ischemia with No Obstructive Coronary Artery (INOCA) and Microvascular Angina (MVA)

**DOI:** 10.3390/jcm12030819

**Published:** 2023-01-19

**Authors:** Vincenzo Sucato, Giuseppina Novo, Cristina Madaudo, Luca Di Fazio, Giuseppe Vadalà, Nicola Caronna, Alessandro D’Agostino, Salvatore Evola, Antonino Tuttolomondo, Alfredo Ruggero Galassi

**Affiliations:** Department of Health Promotion, Mother and Child Care, Internal Medicine and Medical Specialties (ProMISE), University Hospital Paolo Giaccone, University of Palermo, 90127 Palermo, Italy

**Keywords:** microvascular dysfunction, INOCA, global longitudinal strain, microvascular angina, echocardiography

## Abstract

**Background:** The aim of the study is to evaluate the subclinical alterations of cardiac mechanics detected using speckle-tracking echocardiography and compare these data with the coronary angiography indices used during coronary angiography in a population of patients diagnosed with ischemia with no obstructive coronary artery (INOCA) and microvascular angina (MVA). **Methods:** The study included 85 patients admitted to our center between November 2019 and January 2022 who were diagnosed with INOCA compared with a control group of 70 healthy patients. A collection of anamnestic data and a complete cardiovascular physical examination, and echocardiogram at rest with longitudinal strain were performed for all patients. Furthermore, the TIMI frame count (TFC) for the three coronary vessels was calculated according to Gibson’s indications. All parameters were compared with a control population with similar characteristics. **Results:** Patients with INOCA compared to the control population showed statistically significant changes in the parameters assessed on the longitudinal strain analysis. In particular, patients with INOCA showed statistically significant changes in GLS (−16.71) compared to the control population (−19.64) (*p* = 0.003). In patients with INOCA, the total TIMI frame count (tTFC) correlated with the GLS value with a correlation coefficient of 0.418 (*p* = 0.021). **Conclusions:** In patients with angina, documented myocardial ischemia, the absence of angiographically significant stenosis (INOCA) and LVEF > 50%, the prevalence of microvascular dysfunction documented by TFC was extremely represented. A statistically significant reduction in GLS was observed in these patients. TFC and longitudinal strain, therefore, appear to be two reliable, sensitive and easily accessible methods for the study of alterations in coronary microcirculation and the characterization of patients with INOCA and microvascular angina.

## 1. Introduction

The acronym INOCA (ischemia with non-obstructive coronary artery disease) includes any condition responsible for myocardial ischemia in the absence of angiographically significant coronary lesions (stenosis <50%) after performing the coronary angiography [1]. According to some documented cases, up to 40% of patients undergoing coronary angiography for chest pain with anginal features have angiographically normal or nearly normal coronary arteries. INOCA includes a very broad spectrum of physiopathological or clinical conditions capable of determining ischemia, which must always be investigated when approaching the problem. Among these conditions, we must undoubtedly include functional dysfunction of the coronary microcirculation (microvascular angina) [2] that is extremely widespread, 20–30% of patients, mostly women, undergoing coronary angiography for stable angina have a free coronary tree and a microcirculatory disease [3,4]. This condition can be defined as primary microvascular angina (MVA), to distinguish it from forms of MVA in which coronary microvascular dysfunction (CMD) is linked to the presence of specific diseases [5,6,7]. As for the pathogenesis, there are still many obscure points, but the evidence is emerging that underlines the importance of some risk factors over others, as well as evidence of new alterations in endothelial function. This study, therefore, aims to evaluate the subclinical alterations of cardiac mechanics detected using 2D speckle-tracking echocardiography (2D-STE) by comparing the echocardiographic data with the coronary angiography indices used during the execution of coronary angiography in a population of patients diagnosed with INOCA and MVA compared to a control population. Finally, we investigated the correlation between the degree of microvascular dysfunction, assessed with a simple index, such as the TIMI frame count, and the possible extent of the alteration of the global and regional longitudinal strain in patients with INOCA and MVA.

## 2. Materials and Methods

The study included 85 patients (36 men and 49 women) admitted to the Division of Cardiology of the University Hospital Paolo Giaccone in Palermo between November 2019 and January 2022 for signs and symptoms of myocardial ischemia and which showed coronary arteries that do not have angiographically significant stenosis (stenosis less than 50% of the vessel lumen). INOCA was diagnosed in these patients. More specifically, these patients were aged between 35 and 75 and hospitalized for angina or the clinical equivalent of angina (dyspnea and reduced exercise tolerance) who had previously undergone a positive stress test (electrocardiogram from exercise on a treadmill, perfusion myocardial scintigraphy and stress echocardiogram) or who had shown changes on the resting electrocardiogram suggesting myocardial ischemia. All included patients had a preserved left ventricular systolic function (LVEF > 55%) and an absence of evident and significant changes in resting segmental motion. Patients were excluded from the study if documented myocardial ischemia was attributable to non-coronary diseases and determined, for example, by a discrepancy between O_2_ requirement and supply (anemia, COPD, recent onset of uncontrolled hypertension, arteriovenous shunts in the systemic circulation or intracardiacs) or attributable to coronary pathologies immediately identifiable on coronary angiography (coronary anomalies, intramyocardial bridge and multiple intermediate stenosis in series). Patients with moderate and severe valvulopathies, myocardial diseases, and patients with atrioventricular, intraventricular conduction disturbances or with PM/ICD were also excluded due to the consequent intrinsic changes in myocardial kinetics. All patients were contacted by telephone and invited to participate in a follow-up that included a clinical interview, physical examination, resting echocardiogram and myocardial strain analysis using 2D-STE. During the clinical assessment, the risk factors and the characteristics of anginal symptoms that may still be present were investigated through a four-degree scale such as that proposed by the Canadian Cardiovascular Society (CCS). Symptoms and signs typical of heart failure with preserved ejection fraction (dyspnea, asthenia, declining edema and pulmonary stasis) were also carefully investigated. The appearance of new major cardiovascular events after hospitalization (MACE: myocardial infarction, cerebral stroke and new hospitalizations for chest pain) was evaluated. Information was collected on the prescribed home therapy and in some cases it was optimized (for example, with the inclusion of Ranolazine for more effective angina control). All patients underwent blood pressure and heart-rate measurements and a cardiovascular physical examination with particular attention paid to any signs of clinical congestion. The resting echocardiogram was then performed using a General Electric model Vivid q echocardiograph undergoing electrocardiographic monitoring. The main morphofunctional parameters, cardiac kinetics, and diastolic function were evaluated with the use of TDI (Tissue Doppler Imaging) also for the calculation of the E/average e’ ratio. In pall patients, longitudinal strain analysis was performed using the software supplied with the echocardiograph. The images were acquired at an acquisition speed between 50.3 and 54.9 fps using the three analysis projections provided by the software: apical long axis, apical four chambers and apical two chambers. The identification of the AVC (aortic valve closure) was carried out automatically by the software while the definition of the endocardial border and the ROI (region of interest) was manually optimized. The longitudinal strain curves and the global systolic peak strain were then obtained for each of the 18 segments graphically distributed on the bull’s eye. The echocardiographic data of patients with a diagnosis of microvascular angina (positive stress test and normal coronary artery) were compared with those collected in a sample of control patients. This sample consisted of 70 healthy subjects (40 men, 30 women, aged between 35 and 75) with a cardiovascular risk < 5% according to the EURO risk score, with a low probability of unknown myocardial disease, selected from a population that participated in a cardiovascular screening program at the University Hospital Paolo Giaccone in Palermo in the year 2019–2022. In order to evaluate any microvascular dysfunction, angiographic images of all patients were examined. Coronary slow-flow phenomenon (CSFP) is an angiographic phenomenon characterized by the slow passage of contrast in the absence of obstructive coronary artery disease. The diagnosis of the CSFP can be made with a corrected TIMI frame count > 25 frames [8]. Additionally, the TIMI frame count for the three coronary vessels was calculated according to Gibson’s indications, taking care to divide the value obtained for the left anterior descending for a correction factor of 1.7. The possible correlation between the clinical characteristics of the patients, echocardiographic parameters, strain analysis and TIMI frame count was therefore investigated. Statistical analysis was performed using the MedCalc program. The Student’s *t*-test was used to define the statistical significance of the differences between continuous variables between study and control patients and within subgroups of the study population; a *p* value < 0.05 was considered indicative of statistical significance.

## 3. Results

The study population consisted of 85 patients with a diagnosis of INOCA compared with a control group of 70 healthy patients. All patients took part in regular follow ups. At the time of admission, patients with INOCA had a recent history of pain typical of angor, and the remaining patients had worsening dyspnea or asthenia. The mean degree of angina at hospitalization according to the CCS scale was II. In some patients, myocardial ischemia had been documented using more than one diagnostic test. To be specific, the diagnostic tests used were: exercise ECG (41%), resting ECG with transient alterations (ST segment alterations and appearance of negative T waves) (32%), myocardial scintigraphy (19%), resting echocardiogram with transient kinetic changes (6%), and stress echocardiogram (2%). Patients with INOCA and the control group had the same distribution of cardiovascular risk factors. The echocardiogram at rest was carried out in all patients and echocardiographic data was collected in all patients who had a sufficiently adequate acoustic window. Kinetics were normal in all patients undergoing echocardiography except for one patient who presented with apical and periapical segment hypokinesia.

The mean LVEF was 56.15%. Despite the high prevalence of arterial hypertension, the average thickness of the interventricular septum (IVS) was normal (9.72 mm) and only three patients had signs of hypertensive heart disease (IVS ≥ 12 mm). The mean volumes of the left ventricle and left atrium were also normal. Grade I diastolic dysfunction was found in general with an average E/A ratio of 0.95. The average deceleration time was 219.8 ms. The mean lateral TDI s’ was equal to 0.09 m/s (expression of a preserved systolic function of the left ventricle). The average E/e’ ratio was found to be at the upper limits (8.01). The data obtained at the baseline echocardiogram were compared with the control population consisting of 70 healthy patients (who had no history of myocardial ischemia or typical chest pain) with a mean age of 56.65 years old with a profile of a similar risk to the INOCA population. As can be seen, patients with INOCA compared to the control population did not have statistically significant changes in the parameters assessed on the basic echocardiogram. Only a reduction in the values of e’ and a’ at the TDI was observed; however, these values were still within the normal range (Table 1).

The mean global longitudinal strain of the left ventricle (GLS) was equal to −16.7 with an average FPS of 52.23 Hz. Patients with INOCA have a reduced GLS (−16.71) compared to the control population (−19.64) reaching statistical significance (*p* = 0.003) (Table 2).

Furthermore, in patients with INOCA and MVA undergoing coronary angiography, the TFC was evaluated on all three coronary vessels and by performing the sum of the TFCs on all three coronary vessels and obtaining the total TIMI frame count (tTFC). A statistically significant correlation (*p* = 0.021) was observed between the GLS value and the corresponding tTFC value with a correlation coefficient of 0.418 as observed in Figure 1.

The correlation between the strains of the 18 myocardial segments with the TFC of the tributary vessel in the total population with INOCA was evaluated. A statistically significant correlation was observed between the longitudinal strain of the perfused segments from the right coronary artery and TFC of the right coronary artery which was the highest among the TFCs of the three coronary vessels.

## 4. Discussion

The study allowed us to make interesting observations on a complex and elusive pathology which is coronary microvascular dysfunction in patients suffering from microvascular angina.

We wanted to study a paradoxical phenomenon that contrasts with the fundamental corollary of ischemic heart disease, which presupposes the presence of epicardial coronary stenosis as a pathophysiological determinant of ischemia.

From our point of view, investigating the characteristics of this type of patient, often identified as suffering from Cardiac Syndrome X or more recently included in the INOCA category, represented an attempt to better understand the pathophysiology of ischemic heart disease.

It is likely that the problem of ischemia not resulting from epicardial coronary stenosis is actually much more widespread and that it often accompanies the history of obstructive atherosclerotic coronary artery disease with an independent contribution, making its identification even more complex.

From our analysis, 79.4% of patients after revision of the coronarographic clips, presented a slowing of the coronary flow quantified using the TIMI frame count (TFC), which is an indirect expression of microvascular dysfunction.

The high prevalence of microvascular dysfunction that was observed is in line with data from the WISE study and the Sara JD and Widmer RJ studies [9,10] in which the prevalence of microvascular dysfunction assessed with the gold standard of intracoronary Doppler exceeded 60% in the population with angina and healthy coronary arteries without significant differences between the two sexes [1].

The clinical follow-up of our sample of patients also presented data similar to those identified in the WISE study in terms of events and symptoms. As in the WISE study, no significant incidence of major cardiovascular events was observed, although these were still more frequent than in the healthy population. However, similar to the WISE data, there was a high incidence of new hospitalizations for chest pain (occurring in 7.6% of patients) and a significant persistence of angina disorder (in 35.9% of patients) with an important impact on the quality of life of patients [11].

Our study also showed that the severity of anginal symptoms was closely related to the degree of microvascular dysfunction assessed with TFC.

Patients with moderate-to-severe angina (CCS II-IV, equal to 38.4% of the population) had a significantly higher tTFC than the mean value found in the total sample. An observation of this type can only confirm the hypothesis of impaired microcirculation as a fundamental pathogenetic moment in angina with healthy coronaries, relegating the neurogenic hypothesis of the disorder to a secondary level.

Thus, microvascular dysfunction is highly prevalent among patients with healthy coronary ischemia once other easily identifiable causes of ischemia (e.g., coronary discrepancy and abnormalities) have been ruled out. These patients, therefore, do not represent an enigma or a paradox in the context of ischemic heart disease and cannot be considered healthy, but rather must be correctly identified as suffering from microvascular angina and treated accordingly.

From this work, the role of tTFC emerged as a useful marker of microvascular dysfunction and strongly correlates both with the clinical characteristics of patients and with more innovative instrumental indices such as GLS [12]. In patients who instead have obstructive atherosclerotic coronary arteries, microvascular dysfunction could explain the frequent finding of the persistence of angina in patients undergoing effective coronary angioplasty [13].

In our study, in accordance with current scientific evidence [14,15], basal echocardiogram data does not allow the identification of significant changes between the INOCA population and the control population.

Literature data show differences in GLS observed between patients with INOCA or microvascular angina and the control population were statistically significant with mean GLS values of −17.7 ± 2.5% in the Syndrome X population and −15.4 ± 3.3% in the population with microvascular angina (with CFR assessed microvascular dysfunction) and type 2 diabetes mellitus [16,17,18,19,20]. However, larger studies have not shown significant changes in baseline GLS in patients with CFR-assessed microvascular dysfunction, while a statistically significant variation in ∆ GLS has been documented [21].

Baseline GLS was therefore significantly reduced in the population with microvascular angina when microvascular dysfunction was assessed by TFC. The greater the reduction, the more the tTFC is increased, thus revealing an important correlation between GLS and TFC.

Indeed, some studies have documented a close correlation between increased TFC and a reduction in strain rate or longitudinal strain evaluated using 3D-STE [22,23,24].

Therefore, the analysis of the longitudinal strain of the individual segments (rather than the regional one) was found to be related to the TFC of the tributary vessel.

The highest TFC was found for the right coronary artery while the statistically significant alteration of the strain was found in the area of the circumflex artery. However, in the territory perfused by the circumflex artery, by analyzing the longitudinal strain of the single segments, statistically significant alterations were found in the infero-lateral segments. The latter, although commonly identified as territory perfused by the circumflex artery, often represents a border area and overlap between the perfusion dependent on the right coronary artery and that of the circumflex artery.

Likewise, the partial involvement of the apical segments can correlate with the presence of a dominant right coronary artery.

The microvascular dysfunction and the consequent alteration of the TFC probably determines a reduction in myocardial performance mainly in the subendocardial layers which are those whose alteration is most identifiable in the analysis of the longitudinal strain as a consequence of the helical and longitudinal course of the subendocardial fibers.

## 5. Limitations of the Study

There is a lack of evidence in this research field. Although the scientific community are trying to find answers, even today, many pathogenetic mechanisms are not yet clear. Among the limits of the study, it is certainly right to include the number of the sample. The inclusion criteria were very stringent to identify an ideal population of patients with healthy coronary ischemia that could easily be compared with a control population. This made patient selection difficult and justifies the small sample size of the study. Studies of larger populations are needed for more accurate data.

## 6. Conclusions

Ultimately, the documented myocardial ischemia identified on exercise ECG or myocardial scintigraphy that has not been confirmed in terms of significant stenosis on coronary angiography so as to consider the results of the previous false positive tests, finds instead, an instrumental confirmation in 2D strain analysis. Alterations in coronary microcirculation also appear to be capable of causing myocardial ischemia, as well as epicardial coronary stenosis and TFC and longitudinal strain have proved to be two valid tools for the study of those fine alterations of the coronary tree not immediately visible and documented by coronary angiography.

In patients with angina, documented myocardial ischemia, absence of angiographically significant stenosis (INOCA) and LVEF > 50%, the prevalence of microvascular dysfunction documented by TFC was highly represented. A statistically significant reduction in GLS was observed in these patients (−16.77 ± 2.82 vs. −19.64 ± 1.91 with a *p* value of 0.003).

TFC and longitudinal strain, therefore, appear to be two reliable, sensitive and easily accessible methods for the study of alterations in coronary microcirculation and the characterization of patients with INOCA and with microvascular angina. Further studies with larger populations are also necessary to be able to include parameters such as GLS and tTFC in the diagnosis of INOCA and microvascular angina.

## Figures and Tables

**Figure 1 jcm-12-00819-f001:**
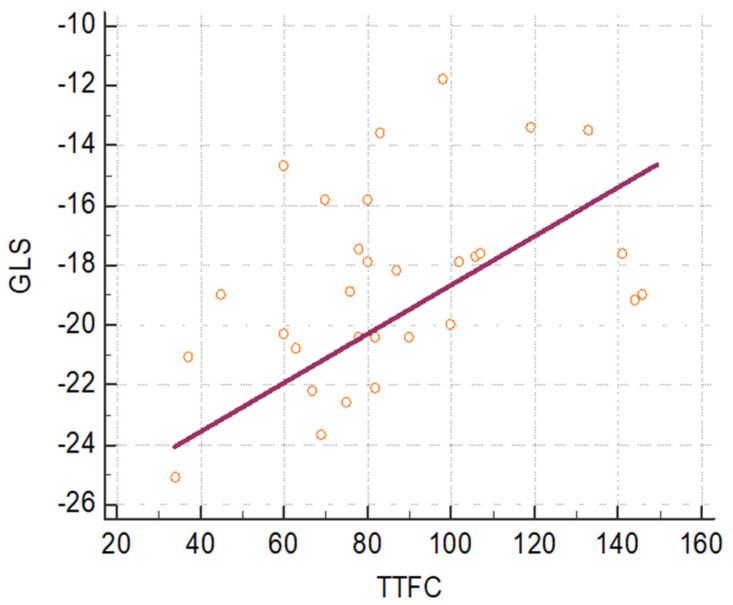
Scatter plot showing a statistically significant correlation (*p* = 0.021) between GLS and tTFC of the population with INOCA (ischemia with non-obstructive coronary artery disease) and MVA (MicroVascular Angina).

**Table 1 jcm-12-00819-t001:** Comparison of baseline echocardiographic data in the INOCA and MVA and control group.

Comparison of Baseline Echocardiographic Data in the INOCA and MVA and Control Group (Mean Values = M and Standard Deviation = SD)					
	INOCA and MVA Group (*n* = 85)		Control Group (*n* = 70)		*p*-Value
	M	SD	M	SD	
Left Ventricular Ejection Fraction (%)	56.15	5.34	58.55	5.7	0.13
Inter Ventricular Septum (mm)	9.72	1.51	9.45	1.15	0.5
End-diastolic diameter (mm)	45.66	5.4	43.1	4.13	0.08
End-diastolic volume (mL)	86.71	19.31	96.8	20.59	0.08
Left atrial volume (mL)	50.20	27.4	44.20	9.59	0.35
E wave (m/s)	0.64	0.16	0.66	0.16	0.66
A wave (m/s)	0.69	0.27	0.60	0.16	0.18
E/A ratio	0.95	0.30	1.13	0.41	0.07
Deceleration time (ms)	219.81	61.32	197.90	41.32	0.16
septal e’ (m/s)	0.07	0.02	0.08	0.03	0.15
septal a’ (m/s)	0.09	0.02	0.11	0.01	f
septal s’ (m/s)	0.07	0.02	0.07	0.01	0.9
lateral e’ (m/s)	0.10	0.03	0.12	0.03	0.02
lateral a’ (m/s)	0.11	0.02	0.13	0.03	0.006
lateral s’ (m/s)	0.09	0.03	0.09	0.02	0.9
E/e’ ratio (m/s)	8.01	4.01	7.34	2.86	0.52

INOCA, ischemia with non-obstructive coronary artery disease; MVA, MicroVascular Angina.

**Table 2 jcm-12-00819-t002:** Comparison of longitudinal strain in patients with INOCA and microvascular angina (AM) and control group.

Longitudinal Strain in INOCA and MVA Patients and Control Group (Mean Values M with Standard Deviation SD)					
	INOCA and MVA Group (*n* = 85)		Control Group (*n* = 70)		*p*-Value
	M	SD	M	SD	
Basal Anterior Septum	−16.8	4.78	−17.85	3.23	0.48
Mid Anterior Septum	−18.6	6.59	−22.6	3.07	0.03
Apical Anterior Septum	−17.9	6.51	−23.3	4.86	0.016
Basal infero-lateral wall	−17.22	4.41	−21.95	2.29	0.0006
Mid infero-lateral wall	−17.5	2.72	−21.35	2.56	0.0007
Apical infero-lateral wall	−18.5	5.24	−20.37	3.64	0.264
Longitudinal strain APLAX (apical long axis view)	−17.22	3.53	−20.69	1.76	0.001
Basal Antero-lateral wall	−18.2	5.77	−19.20	3.59	0.563
Mid Antero-lateral wall	−16.7	6.29	−18.95	2.86	0.185
Apical Antero-lateral wall	−16.22	6.92	−18.55	4.74	0.286
Basal Inferior Septum	−15.2	4.08	−15.84	3.42	0.653
Mid Inferior Septum	−18.4	5.66	−20.20	3.58	0.295
Apical Inferior Septum	−18	6.48	−23.35	4.7	0.015
Longitudinal strain A4C (Apical Four Chamber View)	−16.77	3.71	−18.91	2.35	0.0634
Basal Anterior Wall	−16.3	5.12	−16.40	4.71	0.96
Mid Anterior Wall	−15.2	5.14	−20.80	4.81	0.0065
Apical Anterior Wall	−13.11	7.47	−19.16	5.89	0.022
Basal Inferior Wall	−17.1	3.31	−20.74	3.07	0.006
Mid Inferior Wall	−20.6	3.02	−21.95	4.06	0.204
Apical Inferior Wall	−19	5.39	−21.74	5.66	0.215
Longitudinal strain A2C (Apical Two Chamber View)	−16.8	3.37	−19.50	2.47	0.019
Global Longitudinal Strain (GLS)	−16.77	2.82	−19.64	1.91	0.003

## Data Availability

Not applicable.

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
