# Peer review of "Longitudinal Strain Analysis and Correlation with TIMI Frame Count in Patients with Ischemia with No Obstructive Coronary Artery (INOCA) and Microvascular Angina (MVA)"

_jcm, 2023, doi:10.3390/jcm12030819_

Round 1

Reviewer 1 Report

This is a study concerning the longitudinal strain analysis correlation with TIMI frame count in patients with ischemia with no obstructive coronary artery and microvascular angina.

Although the concept of the study is interesting, there are some important limitations to be reported.

Major comments

1. Did the control group of patients undergo any myocardial ischemia examination? If not, this is an important limitation of your study design as a control group may differ significantly;y for the "active arm" group. How are you sure that the healthy group of people did not have a subclinical myocardial disease? Otherwise, you should refer that you compare people with clinical coronary artery disease vs people without any clinical evidence of coronary artery disease.

2. The manuscript is extremely wordy. Please reduce the introduction and discussion by at least 50%.

3. A limitations paragraph is essential for this manuscript.

4. Do the enrolled patients have elevated troponin?

Minor comments

Abstract

line 13: INOCA and AMV. Please provide the full names (first time in text-abstract)

Introduction

line 35: <50% stenosis does not mean the absence of hemodynamically significant lesions. It means the absence of angiographically significant stenosis.

line 35: angiographic examination. Change to coronary angiography

Results

line 132: mean or median value?

line 167-170: Move this to materials and methods

Author Response

Dear reviewer

Thank you for you revision.

We made all changes. Your suggestions improve the manuscript.

Thanks

Reviewer 2 Report

INOCA and microvascular angina is a hot topic of current research. The authors should answer the following questions.

1.Further clarify the inclusion criteria, especially for non-obstructive coronary artery disease caused by different etiologies, which are highly variable in terms of diagnosis and treatment.

2.Whether coronary flow velocity reserve (CFVR) method can be added for coronary flow evaluation?

3.The value of GLS and tTFC for the diagnosis of INOCA should be further elaborated in the conclusion section

Author Response

Dear Reviewer, thank you for you revision.

We made all changes. Your suggestion improved the manuscript.

Thanks.

Round 2

Reviewer 1 Report

The manuscript has been improved. However, there is still a major concern about the selection of the control group.

Author Response

Dear reviewer

thanks for your comments.

The control group consist of patients with a cardiovascular risk < 5 % according to the EURO risk score. So, there is a low probability of patients with unknow myocardial disease. We compared this group with patients with diagnosis of microvascular angina (positive stress test and normal coronary artery). Both group have a normal echocardiography.

Best regards